# The Regulation of the Growth and Pathogenicity of *Valsa mali* by the Carbon Metabolism Repressor *CreA*

**DOI:** 10.3390/ijms24119252

**Published:** 2023-05-25

**Authors:** Jiyang Jin, Yufei Diao, Xiong Xiong, Chengming Yu, Yehan Tian, Chuanrong Li, Huixiang Liu

**Affiliations:** 1Mountain Tai Forest Ecosystem Research Station of State Forestry Administration, Key Laboratory of Ecological Protection and Safety Prevention of the Lower Yellow River, Forestry College, Shandong Agricultural University, Tai’an 271018, China; m18706386553@163.com (J.J.);; 2Shandong Research Center for Forestry Harmful Biological Control Engineering and Technology, College of Plant Protection, Shandong Agricultural University, Tai’an 271018, China

**Keywords:** *Valsa mali*, carbon catabolite repression, *VmCreA*, growth, pathogenicity

## Abstract

Carbon catabolite repression (CCR) is a very important mechanism for efficient use of carbon sources in the environment and is necessary for the regulation of fungal growth, development, and pathogenesis. Although there have been extensive studies conducted regarding this mechanism in fungi, little is yet known about the effects of *CreA* genes on *Valsa mali*. However, based on the results obtained in this study for the identification of the *VmCreA* gene in *V. mali*, it was determined that the gene was expressed at all stages of fungal growth, with self-repression observed at the transcriptional level. Furthermore, the functional analysis results of the gene deletion mutants (Δ*VmCreA*) and complements (CTΔ*VmCreA*) showed that the *VmCreA* gene played an important role in the growth, development, pathogenicity, and carbon source utilization of *V. mali*.

## 1. Introduction

*Valsa mali* is a mycelial fungus. The destructive Valsa canker disease of apples is caused by the ascomycetes fungus *V. mali* [1]. The occurrence of apple tree rot disease is extremely serious worldwide. It has been found that almost all apple-growing areas are affected by apple Valsa canker, particularly those located in Eastern Asia [2,3].

During the colonization and infection process, the pathogenic fungi face various challenges from hostile environments, such as nutrient limitations and stress conditions. However, a carbon catabolite repression (CCR) mechanism could facilitate the fungi’s effective utilization of the surrounding physiological environment. CCR inhibits the expressions of the genes involved in the utilization of other carbon sources when fungi utilize preferred carbon sources, such as glucose [4]. In *Saccaromyces cerevisiae*, a key regulator of CCR is *Mig1*, which inhibits the transcription of the genes encoding *GAL4* and *SUC2* in the nucleus. The loss of glucose transfers *Mig1* phosphorylation from the protein kinase *Snf1* complex to the cytoplasm [5]. *CreA* is a homologue of yeast *Mig1* in filamentous fungi. However, there are some glaring differences in the mechanism of the CCR. For example, CCR in *S. cerevisiae* involves *Mig1* and the complex *Ssn6/Tup1*. However, the role of this complex has not yet been determined in filamentous fungi [6].

Homologs of the *CreA* gene have been studied in several other filamentous fungi, and *CreA* has been found to be essential for the growth of *Aspergillus nidulans* [7]. In addition, *CreA* mutants were observed to reduce spore production and CCR, counteracting repressor effects [8]. *Cre1* is the gene in *Neurospora crassa* that mainly regulates CCR. In previous studies, when the production of conidia was induced by carbon starvation conditions, *Cre1* mutants on different carbon source media grew significantly slower than the wild type [9]. There is also a homolog of the *CreA* gene in *Beauveria bassiana*, in which the protein contains two C2H2-type zinc finger domains. Real-time PCR analysis results have shown that *BbCre1* expressions are regulated by glucose and have self-transcriptional inhibition characteristics [10]. In *Sclerotinia sclerotiorum*, the CCR repressor gene is *Cre1*, which is known to be functionally associated with the *CreA* of *A. nidulans* but cannot replace the function of *Mig1* in *S. cerevisiae* [5,11]. In *Aspergillus flavus*, *CreA* is a major transcriptional repressor gene. The *CreA* mutants and complements, including *AfCreA*, affect growth and development, spore germination, and nucleus formation and also produce less aflatoxin than the wild type and its complement mutants. *AfCreA* also plays a role in inhibiting the enzymes required to utilize secondary carbon sources [12]. In recent studies, it has been observed that *CreA* also demonstrated involvement in the pathogenicity of *Magnaporthe oryzae*, with *MoCreA* required for the asexual development and pathogenicity of rice blast fungus [4].

At present, the research regarding the CCR of filamentous fungi has made some progress. However, the involved regulation mechanism is still largely ambiguous in *V. mali*. Transcriptome sequencing analysis results have shown that the *CreA* gene (*VM1G_03855*) was differentially expressed during the infection stage of *V. mali* [13]. A *CreA* homolog encoding 434 amino acids, referred to as *VmCreA*, was identified in this study. The role of *CreA* in the development, pathogenicity, and utilization of carbon sources was revealed using deletion and complementation methods.

## 2. Results

### 2.1. Identification of the V. mali CreA

The single copy *VmCreA* gene (Accession number: KUI69101.1) has been identified in the genome of *Valsa mali*. The gene sequence is 1305-bp and encodes a protein of 434 amino acids (AA). Two conserved C2H2 zinc finger DNA-binding domains are predicted at 82–104 and 110–134 AA, including one nuclear localization signal (NLS) and two nuclear export signals (NESs), indicating that *VmCreA* possesses a DNA-binding function and could shuttle between the nucleus and cytoplasm (Figure 1A). This study analyzed the homologous sequences of other fungi and discovered that the C2H2 zinc finger was highly conserved in all of the analyzed species (Figure 1B). A phylogenetic tree was constructed revealing that *VmCreA* had a 100% identity with its homologue in the industrially important *Valsa mali* var. *Pyri* (Figure 1C).

### 2.2. Construction of the VmCreA Deletion and Complementation of the Mutant Strains

To confirm whether *VmCreA* participates in the CCR of *V. mali* and explore its biological role in the growth, development, and pathogenicity processes of *V. mali*, *VmCreA* deletion mutants were first constructed in this study using the Double-joint PCR approach. The *VmCreA* was first homologously replaced with a hygromycin B resistance gene (hph) in the genome of the wild-type strain sdau11-175 (WT). The *CreA* ORF region was fused in-frame with the *eGFP* gene for complementation (Figure 2A), and the transformants were verified using diagnostic PCR and qRT-PCR. The diagnostic PCR revealed that the deletion vector had been successfully constructed (Figure 2B). Southern-blot indicates that this mutant strain is a single-copy gene knockout mutant (Appendix A). The RT-PCR results showed the absence and regular transcript levels of the *CreA* in the Δ*CreA* and CTΔ*CreA* strains, respectively (Figure 2C).

### 2.3. Influencing Effects of VmCreA on the Vegetative Growth and Development of V. mali

The results of this study revealed that the deletion of *VmCreA* dramatically affected the colony morphology of *V. mali*. For the wild-type strains, the Δ*VmCreA* mutants and complementary transformant were all cultured on PDA medium. It was observed that the mycelial growth rate of Δ*VmCreA* was slower when compared with the WT and CTΔ*VmCreA* (Figure 3A). In addition, all of the strains were inoculated on minimal media (MM) with different substrates as the only carbon sources, including glucose, lactose, sucrose, starch, carboxymethy cellulose (CMC), and 2-deoxyglucose. Then, after five days, the colony diameter of Δ*VmCreA* was found to be lower. Therefore, based on the above results, it was suggested that *VmCreA* was necessary for the vegetative growth and the utilization of different carbon sources in *V. mali* (Figure 3B). However, the effects of *VmCreA* on fungal spores in *V. mali* could not be determined.

### 2.4. VmCreA Participates in the CCR of V. mali

This study conducted further experiments to verify whether *VmCreA* is involved in the regulation of CCR. The growth-inhibition rates of the WT, Δ*VmCreA*, and CTΔ*VmCreA*, which had been inoculated on MM with CMC were found to be generally consistent. However, when grown on medium with 2-deoxyglucose and CMC as the common carbon sources, Δ*VmCreA* had a growth rate consistent with that achieved with a single carbon source. It was also found that the growth rates of the WT and CTΔ*VmCreA* were significantly inhibited under certain conditions, as shown in Figure 4A. In such cases, the 2-deoxyglucose could be identified but had not been utilized, indicating that the WT utilized neither 2-deoxyglucose nor CMC, and growth was significantly inhibited. Meanwhile, although there was a lack of CCR, Δ*VmCreA* was still able to grow by utilizing CMC on a medium containing 2-deoxyglucose (Figure 4B). It was observed that when starch was used as the only carbon source, none of the strains were stained blue by the iodine liquid, indicating that the strains had all used the starch for growth. However, when both starch and glucose were used as the common carbon sources, the Δ*VmCreA* was not stained blue by the iodine liquid. Those findings indicated that the Δ*VmCreA* has a CCR defect and can use both glucose and starch. Therefore, it could be concluded that the *VmCreA* is involved in regulating the CCR of *V. mali* (Figure 4C).

### 2.5. Stress Responses of the V. mali Are Influenced by VmCreA

This study observed the responses of the strains to osmotic and cell wall stress agents in order to determine the influencing effects of *VmCreA* on the regulation responses of the *V. mali* for cell wall homeostasis. The mycelial growth rates of the WT, Δ*VmCreA*, and CTΔ*VmCreA* were all inhibited under the osmotic stress. However, the Δ*VmCreA* was found to be more sensitive to the cell wall stress induced by CFW and SDS than the WT and CTΔ*VmCreA* strains. Therefore, based on those findings, it was concluded that the *VmCreA* had influenced the responses of the *V. mali* to cell wall stress agents (Figure 5).

### 2.6. VmCreA Is Required for V. mali Pathogenicity

Mycelial plugs of the various strains were inoculated onto apple twigs and fruit. Then, following a culturing period of seven days, it was observed that the lengths of the lesions on the apple fruit with Δ*VmCreA* inoculations had decreased by approximately 53% compared to those with the WT (Figure 6A). It was also found that the lengths of the lesions on the apple twigs were reduced by approximately 57% (Figure 6B). In addition, the pathogenicity of the complement mutants had returned to the original levels. The above results indicated that the *VmCreA* is required for *V. mali* pathogenicity.

### 2.7. VmCreA Regulates Hydrolases Production

In order to further understand the role of *VmCreA* in the synthesis of the *V. mali* cell wall-hydrolase, the expressions of 13 genes were detected using qRT-PCR. It was discovered that the expressions of 12 genes in the Δ*VmCreA* mutant were inhibited when compared to the WT. The inhibited genes included *VM1G_02565* (EndopolygalacturonaseD), *VM1G_03512* (Glucoamylase), and *VM1G_10963* (Pectin lyase B). However, the *VM1G_03115* encoding of Aspartic protease was found to be significantly up-regulated in the Δ*VmCreA* mutant, as shown in Figure 7.

## 3. Discussion

Carbon catabolite repression (CCR) is widely found in various fungi. CCR is a process used to preferentially utilize preferred carbon sources from the environment. Environmental signals can stimulate CCR to regulate the growth, development, and pathogenic abilities of fungi [16]. The Valsa canker disease in apples caused by *V. mali* is an important biological disaster for apple crops. At present, there appears to be a lack of systematic research on the role of CCR in the pathogenicity of this type of fungus.

In this study, the experimental results demonstrated that *VmCreA* is a key regulator of *V. mali* CCR. CreA is a protein containing two C2H2-type zinc finger domains, and its zinc finger DNA-binding domain is highly homologous to that of the stress-regulator genes *Mig1* in *S. cerevisiae* [11]. When glucose was limited by *S. cerevisiae*, *Mig1* promotes the genes that encode the enzymes that utilize other secondary carbon sources to relieve CCR [17]. However, the glucose repressor *CRE1* from *S. sclerotiorum* is functionally related to CREA from *A. nidulans* but not to the Mig proteins from *S. cerevisiae* [18]. In filamentous fungi, *CreA* has been identified as the key regulator of the carbon catabolite repression (CCR) responsible for efficiently utilizing carbon sources in the environment that can bind to the conserved 5′-SYGGRG-3′ on the promoter region of related genes [19]. In previous studies involving transcriptome sequencing analyses, it was found that the *CreA* gene (*VM1G_03855*) was up-regulated 2.5-fold (12 days versus 3 days) during the hyphal growth stage [13]. Since CCR-binding sites (CCCGC and CTCCAG) were found in the upstream of the *VmCreA*, it was inferred that the *VmCreA* may have self-inhibitory effects at certain transcription levels. This study conducted analyses of the relative expression levels of *VmCreA* at different glucose induction times using qRT-PCR. The results showed that *VmCreA* expression was relatively high within one hour of the glucose addition and had significantly decreased thereafter. Therefore, based on the analysis results, it could be inferred that the *VmCreA* expression was inhibited by the glucose. In other words, there may have been self-inhibitory effects (Appendix A).

It has been observed that *CreA* is relatively conserved in filamentous fungi. Sequence alignments have shown some highly conserved regions throughout various species, and the *V. mali* also contained the typical CCR nuclear penetration signal “PNSRRG”. It has also been confirmed that *MoCreA* is mainly localized in the nucleus [4]. Therefore, it was necessary in this study to verify whether the same features also applied to the *VmCreA* in *V. mali*. Mycelia basically give fungi their shape and size characteristics. This study discovered that the growth of Δ*CreA* mycelium was impaired, and its growth rate was significantly slowed down on MM with different substrates as the only carbon sources. The reduced number of mycelia indicated that the *CreA* was required for the growth of the *V. mali*. The growth of the Δ*VmCreA* was inhibited by CMC, starch, and lactose, indicating that *VmCreA* can regulate the expression of related enzymes using these carbon sources, and its knockdown affects the utilization of these carbon sources; further studies should confirm this aspect. However, it was reported that the disruption of *CreA* in *T. reesei* inhibits hyphal growth and leads to a decrease in cellulase production [20]. Of further interest is a deletion removing the *CreA* gene in *A. nidulans*, which allows limited germination of the spore but not colony formation; this result is the opposite of the result in the present study [8].

Osmotic stress has been shown to positively affect the morphology and conidiation of Δ*CreA* in *A. flavus* [21]. However, this was not found to be the case in *V. mali*. The growth of Δ*CreA* was significantly impaired under the cell wall stress conditions induced by SDS and CFW, which was similar to the observations in *A. flavus* [12], where evidence was found that *CreA* could be involved in cell wall integrity.

Hydrolases, such as pectinases, cutinases, and amylases, play important roles in fungal virulence, as they catalyze the production of secondary metabolites [22]. In this study, the hydrolases exhibited dramatic decreases in transcription in the Δ*VmCreA* mutant, and the down-regulated expressions of the hydrolase genes may have negatively affected the *V. mali* virulence.

The infection process of Valsa canker disease in apples is a rather complex interaction process that includes cell wall hydrolases, small molecules such as toxins, and effector proteins [23]. This study’s results revealed that *VmCreA* influenced the morphology, vegetative growth, stress responses, and pathogenicity of the *V. mali*. This is the first report regarding the role of *CreA* in *V. mali*. However, the regulation mechanism of *VmCreA* remains limited, with more scientific data still required to verify the findings of this research.

## 4. Materials and Methods

### 4.1. Fungal Strains and Culture Manipulations

*Valsa mali* (Vm) isolate sdau11-175 was used as the wild-type control strain throughout this study (sdau11-175 was collected from Yantai by the Shandong Research Center for Forestry Harmful Biological Control Engineering and Technology in 2011, and it is a high-Pathogenicity variety), and the null mutants were genetically derived from sdau11-175. All of the strains were maintained at 25 °C on potato dextrose agar (PDA). Cultures for the protoplast preparation were cultivated in liquid potato dextrose (PD) and then incubated in a conical bottle at 25 °C and 120 rpm for three days. PEG-mediated transformation of the protoplast was performed to obtain the knockout mutants. The media used in this study included minimal medium (MM); complete medium (CM); and TB3 medium. For testing the utilization of the different carbon sources, the strains were grown on minimal medium (MM) with 2% (*W*/*V*) glucose or an equivalent amount of lactose, sucrose, starch, carboxymethy cellulose (CMC), and 2-deoxyglucose, respectively, as the only carbon sources. For this study’s investigation of whether *VmCreA* is involved in the CCR pathways, the strains were grown on MM containing 2% (*W*/*V*) glucose, with or without 1% (*W*/*V*) starch.

Each of this study’s experiments contained three replicates for each strain, and all of the experiments were carried out three times.

### 4.2. Cloning and Sequencing Analysis of the VmCreA

The *VmCreA* (KUI69101.1) sequence was downloaded from the National Center for Biotechnology Information Resources (NCBI). NCBI Conserved Domain Search (https://www.ncbi.nlm.nih.gov/cdd, accessed on 19 May 2021); Signal-5.0 (https://services.healthtech.dtu.dk/service.php?SignalP-5.0, accessed on 22 May 2021); and Pfam (http://pfam.xfam.org/, accessed on 22 May 2021) online analysis software were utilized to analyze the domain and predict the signal peptide of the amino acid sequence encoded by *VmCreA*. Then, the CreA protein sequences of *Valsa mali* var. *Pyri* (KUI53877.1); *Diaporthe helianthi* (POS81118.1); *Neurospora crassa* (XP_961994.1); *Daldinia childiae* (XP_033439606.1); *Fusarium bulbicola* (KAF5969685.1); *Pyricularia oryzae* (XP_003714427.1); *Saccharomyces cerevisiae* (AAT93178.1); *Colletotrichum higginsianum* (CCF41975.1); *Fusarium oxysporum* (SCO85024.1); and *Beauveria bassiana* (ABL07611.1) were aligned by ClustalW using MEGA 6 software. A neighbor-joining phylogenetic tree was also constructed.

### 4.3. Targeted Deletion and Complementation of the CreA Gene

In this study, the *CreA* deletion mutant strain was created using Double-joint PCR [24]. Two flanking fragments of the ORF of *VmCreA* were amplified using the primer pairs *CreA-UF/CreA-UR* and *CreA-DF/CreA-DR*, respectively, and fused to either side of hygromycin B phosphotransferase genes (*hph*) that were amplified using the primer pairs *hph-F/hph-R* using Double-joint PCR. The PCR products were co-transformed into protoplasts of the wild-type strain sdau11-175. The positive transformants were verified using diagnostic PCR with the primers *YZ-F1/YZ-R1*, *YZ-F2/YZ-R2*, and *CreA-F/CreA-R* (Appendix A) for *VmCreA* ORF upstream fragments (containing the 5′ flanking sequence of the *CreA* gene and a part of the *hph*) and downstream fragments (containing a part of the *hph* and the 3′ flanking sequence of the *CreA* gene), respectively.

For the complementation, a 2.36 kb genomic DNA fragment containing the native promoter and ORF region was amplified using the primer pair *HB-F/HB-R* (Appendix A), cloned into the vector pYF11 containing the hygromycin B gene, and then fused in-frame with the *eGFP* gene. The resulting complementation vector was reintroduced into the Δ*VmCreA* mutant through protoplast transformation. The complementary transformants were first screened using hygromycin and then confirmed using qRT-PCR.

### 4.4. Quantitative Reverse Transcription Polymerase Chain Reactions

In this study, the *G6PDH* (Glyceraldehyde-6-phosphate dehydrogenase) gene was used as an internal control [25]. The mycelia were harvested at 3, 6, and 9 days post-inoculation on PDA medium from all the examined strains. The total RNA was extracted using a TRIzol reagent, and the first strand cDNA was synthesized using a Prime Script TM RT reagent kit with gDNA Eraser (Perfect Real Time) kits. Subsequently, qRT-PCR was performed using a Bio-Rad CFX-96 PCR instrument. The relative expressions of the transcripts were calculated using the 2^−ΔΔCT^ method [26]. The primers for qRT-PCR used in this study are listed in Appendix A.

### 4.5. Observations of the Growth Rates, Conidia, and Morphological Processes

The PDA medium samples were point inoculated with 5 mm of fungus to evaluate the growth rates. The plates were incubated at 25 °C for three days. The conidia production was calculated according to the amount of pycnidia and the growth rates of the wild-type and mutant strains on PDA medium. The samples had been cultured on PDA medium at 25 °C under dark and light conditions for 25 days [27].

### 4.6. Cell-Wall Integrity Inhibitor Stress Assay

In the present study, PDA medium containing different stress agents was inoculated with 5 mm fungus of wild-type and mutant strains and then incubated at 25 °C for four days. The stress agents included osmotic stress NaCl (sodium chloride, 200 ppm) and Sor (Sorbitol, 1 M); cell wall stress SDS (sodium dodecyl sulphate, 0.01%); CFW (Calcofluor White, 200 ppm); and CR (Congo Red, 300 ppm).

### 4.7. Pathogenic Determination Assays

For assaying the virulence on the pathogenic hosts, wild-type and deletion mutants were cultured on PDA medium for four days. Then, 5 mm agar plugs were taken from the edge of a colony and inoculated via prick wounds on the fruit and one-year old twigs of *Malus domestica* “Fuji”. The inoculated tissues were incubated at 25 °C for seven days, and the lengths of the lesions were recorded using a crossing method.

### 4.8. Statistical Analysis

Statistical and significance analyses were carried out using SPSS software (IBM SPSS Statistics 20), and significance was recognized if the *p*-values were <0.05. All the assay results were differentiated using a one-way analysis of variance method.
inhibition rate (%) = [(dc **−** dt)/dc] × 100

## 5. Conclusions

In summary, we constructed *VmCreA* deletion mutants and found that *VmCreA* plays crucial roles in the growth, development, pathogenicity, and carbon source utilization. Our research provides clear evidence that testifies to the molecular pathogenic mechanism of *VmCreA* in *V. mali*.

## Figures and Tables

**Figure 1 ijms-24-09252-f001:**
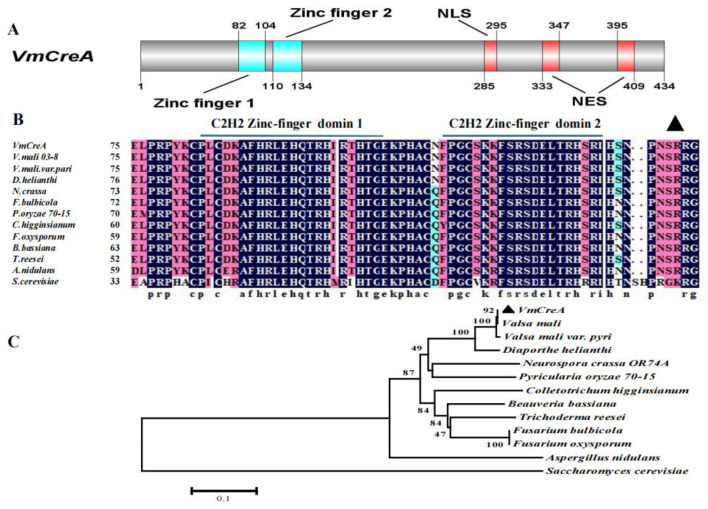
Domain prediction, sequence alignment, and phylogenetic analysis of *VmCreA*: (**A**) Domain predictions were performed using Pfam, and nuclear localization signal (NLS) and nuclear export signal (NES) predictions were performed using cNLS Mapper [14]. A schematic diagram of the protein domains and functional motifs was drawn using DOG 2.0 [15]. The numbers denote the corresponding positions of the predicted domains in the protein sequence. (**B**) Sequence alignment of the zinc-finger domains and the conserved phosphorylation site of *VmCreA* with its homologs from several other fungi. (**C**) Phylogenetic relationship of proteins involved in carbon catabolite repression among the *V. mali* and several other fungi. The scale bar corresponds to the numbers of amino acid substitutions per site.

**Figure 2 ijms-24-09252-f002:**
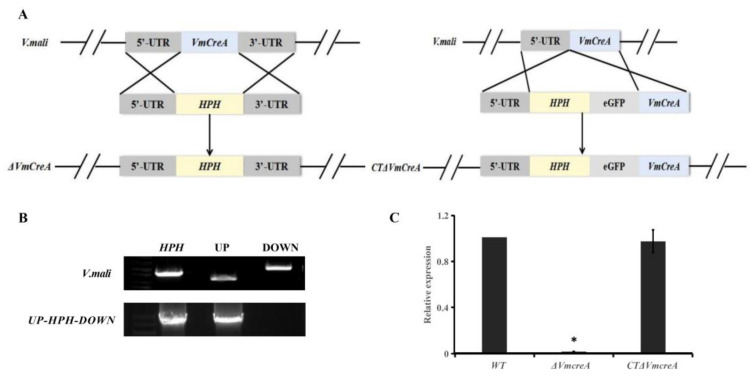
Verification of Δ*VmCreA* and CTΔ*VmCreA* strains: (**A**) diagrammatic representation of the gene replacement strategy for the construction of the Δ*VmCreA* and CTΔ*VmCreA* strains; (**B**) PCR amplification of the segments *HPH*, UP, and DOWN and of the ligation fragments UP-*HPH*-DOWN; (**C**) qRT-PCR verifications of the WT, Δ*VmCreA*, and CTΔ*VmCreA* strains (The single asterisk indicate significant differences at *p* < 0.05).

**Figure 3 ijms-24-09252-f003:**
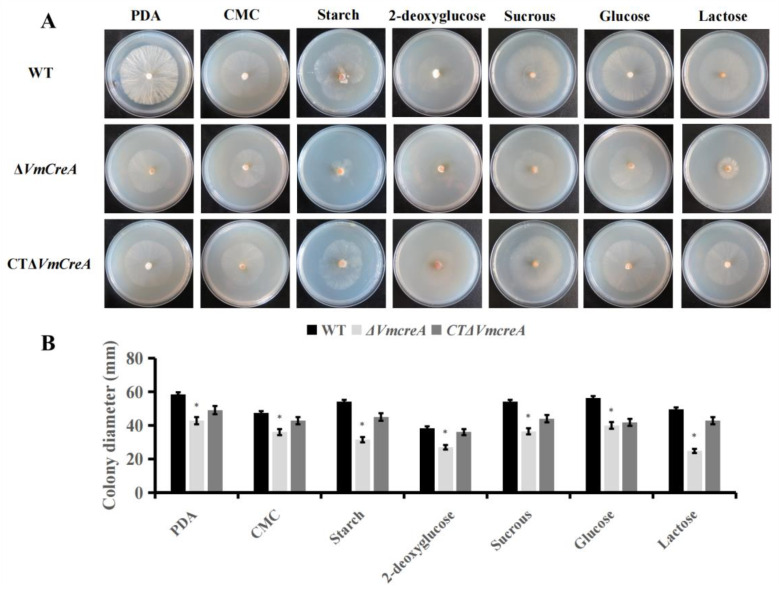
*VmCreA* deletion leads to serious defects in vegetative growth: (**A**) colony morphology of the WT, ΔVm*CreA*, and CTΔ*VmCreA* strains grown on PDA minimal media (MM) containing various carbon sources, including 2% CMC, 2% starch, 2% 2-deoxyglucose, 2% sucrose, 2% glucose, and 2% lactose; (**B**) bar graph representing SD from three independent experiments with three replicates, in which the asterisks indicate statistically significant differences (The single asterisk indicate significant differences at *p* < 0.05).

**Figure 4 ijms-24-09252-f004:**
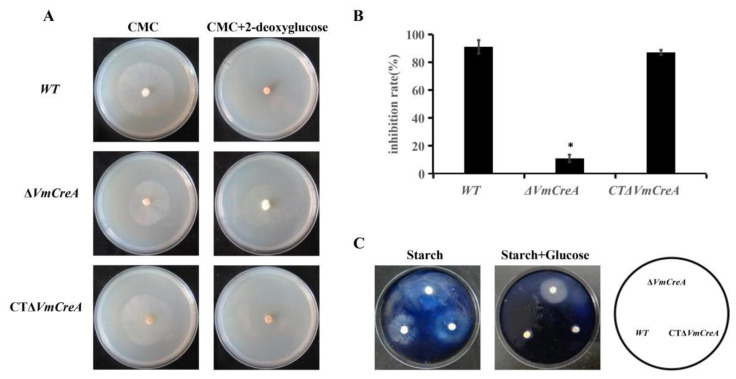
Involvement of *VmCreA* in the regulation of the CCR of *V. mali*: (**A**) mycelia growth of *VmCreA* knockout and complement mutants on minimal media (MM) with CMC and 2-deoxyglucose as the carbon sources; (**B**) inhibition of the growth rates of the strains in the panel (The single asterisk indicate significant differences at *p* < 0.05); (**C**) growth rates with 1% glucose and 1% starch as the carbon sources for three days and the iodine-staining results.

**Figure 5 ijms-24-09252-f005:**
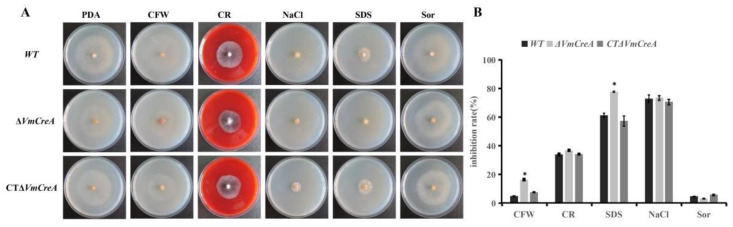
Inhibition of the growth rates of the *V. mali* strains under different stress conditions: (**A**) colony morphology of the WT, ΔVm*CreA*, and CTΔ*VmCreA* strains grown on PDA, or PDA containing 100 μg/mL CFW, 100 μg/mL CR, 0.01% SDS, 0.5 mol/L NaCl, and 1 mol/L at 25 °C for four days, respectively; (**B**) inhibition of the growth rates of the strains in the panel (The single asterisk indicate significant differences at *p* < 0.05).

**Figure 6 ijms-24-09252-f006:**
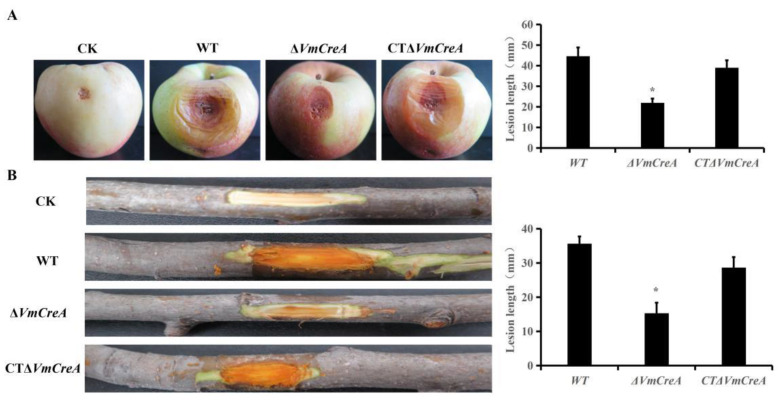
Pathogenicity assay of *V. mali*, where the bars represent the SD from three independent experiments with three replicates (The single asterisk indicate significant differences at *p* < 0.05): (**A**) morphology of the *V. mali* WT, Δ*VmCreA*, and CTΔ*VmCreA* strains on apple fruit after seven days of inoculation; (**B**) morphology of the *V. mali* WT, Δ*VmCreA*, and CTΔ*VmCreA* strains on apple twigs after seven days of inoculation.

**Figure 7 ijms-24-09252-f007:**
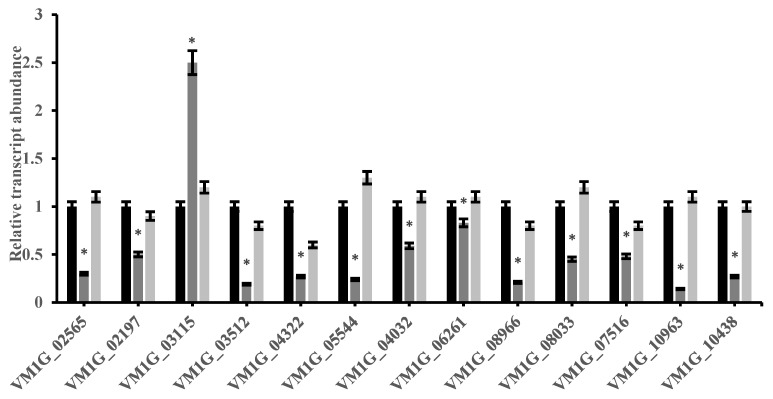
Results of the qRT-PCR analysis of the hydrolase-related genes: *VM1G_02565* (Endopolygalacturonase D); *VM1G_02197* (Endo-xylogalacturonan hydrolase A); *VM1G_03115* (Aspartic protease pep1); *VM1G_03512* (Glucoamylase); *VM1G_04322* (Exopolygalacturonase B); *VM1G_05544* (Cutinase); *VM1G_04032* (Cutinase transcription factor 1 alpha); *VM1G_06261* (Pectin lyase A); *VM1G_08966* (Pectinesterase A): *VM1G_08033* (Endopolygalacturonase D); *VM1G_07516* (Rhamnogalacturonate lyase B); *VM1G_10963* (Pectin lyase B); and *VM1G_10438* (Pectinesterase). (The single asterisk indicate significant differences at *p* < 0.05).

## Data Availability

All the data that support the findings of this study are available in the paper and its Appendix A published online.

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
