# Peer review of "The Regulation of the Growth and Pathogenicity of Valsa mali by the Carbon Metabolism Repressor CreA"

_ijms, 2023, doi:10.3390/ijms24119252_

Round 1
Reviewer 1 Report
This manuscript reported the biological function of CreA in the development and pathogenesis of Valsa mali, and found that CreA negatively regulates the expression of cell wall-hydrolase genes, which is interesting.
Lines 54-57, in the rice blast fungus, two reports found that CreA and Crf1 synergistically regulate lipid catabolism (Cao et al. Characterization of 47 Cys2 -His2 zinc finger proteins required for the development and pathogenicity of the rice blast fungus Magnaporthe oryzae. New Phytol. 2016 211(3):1035-51. doi: 10.1111/nph.13948. PMID: 27041000; Cao et al. The basic helix-loop-helix transcription factor Crf1 is required for development and pathogenicity of the rice blast fungus by regulating carbohydrate and lipid metabolism. Environ Microbiol. 2018 20(9):3427-3441. doi: 10.1111/1462-2920.14387. PMID: 30126031.). This information of CreA should be discussed in the introduction or discussion.
In Figure 2, PCR is required to verify that the VmCREA gene in the mutant has been deleted. The single copy of HPH gene inserted in the mutant also needs to be verified by South Blot or qPCR. And in Figure 2B, the UP-HPH-DOWN fragment is missing.
Figure 3B, because sugars are not inhibitors and do not inhibit the growth of the fungus, it is appropriate to use the relative growth rate here.
Figure 7, the data needs to be analyzed for significant differences.
Reviewer 2 Report
Carbon catabolite repression (CCR) is an important mechanism for regulating fungal growth, development and pathogenicity. In this study, the authors confirmed that the homologous gene VmCreA from Valsa mali plays an important role in growth and development, pathogenicity and carbon source utilization of M. mari by gene knockout. This finding has important implications for the identification of CCR regulators. Here, I offer some suggestions for the author's reference, as described below.
1. In abstract: “. However, based on the results obtained in this study for the identification of the VmCreA gene in V. mali, it was determined that the gene was expressed at all stages of fungal growth, with self-repression observed at the transcriptional level.” However, I did not find the corresponding supporting graph and table in the results section or supplementary files.
2. Please list the specific calculation method of inhibition rate in Materials and methods section
Other comments:
1) Please list the full name of the abbreviations such as WT, CT, etc. when they first appear
2) Please supplement and improve the legend in Figure 3 (B)
3) In Figure 6. Pathogenicity assay of V. mali, where the bars represent the SD from three independent experiments with three replicates (P < 0.05): (A) Morphology of apple fruit inoculated with V. mali WT, ΔVmCreA, and CTΔVmCreA strains for seven days; (B) Morphology of apple twigs inoculated with V. mali WT, ΔVmCreA, and CTΔVmCreA strains for seven days
4) Line 314, please list the version of SPSS software
5) Please indicate statistically significant differences in Figure S1
Reviewer 3 Report
In the manuscript ‘The Regulation of the Growth and Pathogenicity of Valsa mali by the Carbon Metabolism Repressor CreA’ the authors performed the set of experiments, proving that CreA participates in the Carbon Catabolite Repression in apple canker pathogenic fungus Valsa mali. Moreover, the authors performed a set of experiments, proving that CreA is self-regulated on the transcriptional level and play role in pathogenesis against plant host. In my opinion, the presented work is interesting and it is important to the other fungal researchers, working with Valsa mali and other pathogenic fungi. The carefully thought-out experiments gave the interesting results. However, I have some points, which might improve the presented data:
Major remarks:
Since You constructed GFP-CreA protein, it would be good to confirm e.g. its nuclear localization, dual localization, etc (row 71-72). Again, row 214 – please, show GFP-CreA nuclear localization to confirm your assumptions
Figure 1 should contains those CreA, which are described in literature, such as A. nidulans and T. reesei. Three Valsa mali CreA are redundant, one would be sufficient
The deletion should be confirmed by Southern blot, to confirm single integration in-locus and avoid unwanted random-genomic integrations
Figure 2B is unclear; please, rewrite this description to clearly explain the knock-out construction
Figure 3B – the results indicate that the N-terminal GFP-tag affects CreA regulation, leading to its higher de-repression/activity (lower inhibition rate) on the media with de-repressive carbon sources, such as CMC or lactose. It should be discussed and explained.
Row 171: secretion of some proteases under carbon starvation lead to autolysis, hence some experiments should be included to confirm or exclude this process
Row 289: PDA medium is a type of undefined complete medium. To be sure that experiments are repetitive and transparent, the minimal medium should be used to all quantitative experiments. Unless the results are comparable.
Different modes of CreA regulation should be discussed, regarding published data from A. nidulans, T. reesei, S. sclerotiorum and A. chrysogenum
Minor remarks:
Row 31: ‘expression’, inversely – glucose is an easy carbon source, rarely found in nature solely and represses metabolism by CCR. Fungi have no choice and use complex carbon sources. In the presence of glucose (favourable carbon source), CCR represses many genes involved in acquisition of these complex carbon sources to save energy and avoid ATP unnecessary consumption.
Row 40: nidulans
Row 59: obscured – language!
Row 84: Phylogenetic
Row 96: please, explain CTD abbreviation
Row 212: import, instead penetrate
Row 285: hygromycin, instead ampicillin
Reviewer 4 Report
Find the attached file

Round 2
Reviewer 1 Report
Most comments have been revised.
Author Response
非常感谢您的友好工作和对我们论文发表的考虑。我谨代表我的合著者向你表示衷心的感谢。
Reviewer 3 Report
The manuscript ‘The Regulation of the Growth and Pathogenicity of Valsa mali by the Carbon Metabolism Repressor CreA’ corresponds to an updated version of a manuscript previously submitted to International Journal of Molecular Sciences. From the methodical point of view I have no complaints. In this submission the authors have addressed few recommendations to the previous version of the manuscript. The language is improved. However, I still see some points to correct:
Major remarks:
GFP-CreA protein microscopic image may be included as an evidence of cytoplasmic localization. However, since You claimed CreA nuclear localization, it should be proved to confirm your assumptions (probably it depends on carbon source)
Again: Figure 1 should contains those CreA, which are described in literature, such as A. nidulans and T. reesei.
Southern blot analysis may be added to figure 2 or attached as a supplementary file
Again: Figure 3B – the results indicate that the N-terminal GFP-tag affects CreA regulation, leading to its higher de-repression/activity (better growth) on the media with de-repressive carbon sources, such as CMC or lactose. It should be discussed and explained in Discussion section.
Again: growth on PDA medium is a type of undefined complete medium. I did not recommended sugar-free medium as authors misunderstood. To be sure that experiments are repetitive and transparent, the minimal medium with DEFINIED carbon source (such as 1% glucose) should be used to all quantitative experiments. Unless the results are comparable.
Again: Different modes of CreA regulation should be discussed, regarding published data from A. nidulans, T. reesei, S. sclerotiorum and A. chrysogenum
Minor remarks:
Again: Row 33: it is completely in the other way round. Glucose triggers CCR and represses expression of the genes involved in acquisition of 'difficult' carbon sources. The sentence: 'Fungi can preferentially utilize preferred carbon sources by inhibiting the expressions of the genes involved in the utilization of other carbon sources, such as glucose' indicates that the Authors mislead CCR mechanism.
Author Response
Please see the attachment.
Thanks very much for your kind work and consideration on publication of our paper. On behalf of my co-authors, we would like to express our great appreciation to you.

Round 3
Reviewer 3 Report
I have some minor remarks:
row 31: please change sentence order to make sense: CCR inhibits the expressions of the genes involved in the utilization of other carbon sources, when fungi utilize preferred carbon sources, such as glucose.
row 197: Mig1 was phosphorylated by protein - space
197-199 - this sentence lost the sense. please, re-write them, because did not refer to logical sense.
Author Response
Please refer to the attachment. Many thanks for your kind work and consideration of our paper for publication. On behalf of my co-authors, I would like to express my sincere thanks to you.
